# Cloning, Expression, Characterization and Immobilization of a Recombinant Carboxylesterase from the Halophilic Archaeon, *Halobacterium salinarum* NCR-1

**DOI:** 10.3390/biom14050534

**Published:** 2024-04-30

**Authors:** Nestor David Ortega-de la Rosa, Evelyn Romero-Borbón, Jorge Alberto Rodríguez, Angeles Camacho-Ruiz, Jesús Córdova

**Affiliations:** 1Centro Universitario de Tlajomulco, Departamento de Ingeniería Biología, Sintética y de Materiales, Universidad de Guadalajara, Carretera Tlajomulco-Santa Fé Km. 3.5 No.595, Lomas de Tejeda, Tlajomulco de Zúñiga 45641, Mexico; nestor.ortega@academicos.udg.mx; 2Centro Universitario de Ciencias Exactas e Ingenierías, Departamento de Química, Universidad de Guadalajara, Blvd. Gral. Marcelino García Barragán 1421, Col. Olímpica, Guadalajara 44430, Mexico; evelyn.romero@academicos.udg.mx; 3Biotecnología Industrial, Centro de Investigación y Asistencia en Tecnología y Diseño del Estado de Jalisco A. C., Camino el Arenero 1227, El Bajío del arenal, Zapopan 45019, Mexico; jrodriguez@ciatej.mx; 4Centro Universitario del Norte, Departamento de Fundamentos del Conocimiento, Universidad de Guadalajara, Carretera Federal Km. 191 No. 23, Col. Santiago Tlaltelolco, Colotlán 46200, Mexico; angeles_camacho@cunorte.udg.mx

**Keywords:** recombinant carboxylesterase, enzyme immobilization, biochemical characterization, halophilic archaea

## Abstract

Only a few halophilic archaea producing carboxylesterases have been reported. The limited research on biocatalytic characteristics of archaeal esterases is primarily due to their very low production in native organisms. A gene encoding carboxylesterase from *Halobacterium salinarum* NRC-1 was cloned and successfully expressed in *Haloferax volcanii*. The recombinant carboxylesterase (rHsEst) was purified by affinity chromatography with a yield of 81%, and its molecular weight was estimated by SDS-PAGE (33 kDa). The best kinetic parameters of rHsEst were achieved using *p*-nitrophenyl valerate as substrate (K_M_ = 78 µM, k_cat_ = 0.67 s^−1^). rHsEst exhibited great stability to most metal ions tested and some solvents (diethyl ether, *n*-hexane, *n*-heptane). Purified rHsEst was effectively immobilized using Celite 545. Esterase activities of rHsEst were confirmed by substrate specificity studies. The presence of a serine residue in rHsEst active site was revealed through inhibition with PMSF. The pH for optimal activity of free rHsEst was 8, while for immobilized rHsEst, maximal activity was at a pH range between 8 to 10. Immobilization of rHsEst increased its thermostability, halophilicity and protection against inhibitors such as EDTA, BME and PMSF. Remarkably, immobilized rHsEst was stable and active in NaCl concentrations as high as 5M. These biochemical characteristics of immobilized rHsEst reveal its potential as a biocatalyst for industrial applications.

## 1. Introduction

Archaea constitute a domain of prokaryotic organisms found in almost every habitat on the planet. Some have adapted to survive in extreme environmental conditions [1]. These extreme environments are characterized by high (T > 70 °C) or low (T < 0 °C) temperatures, high alkalinity (pH > 10) or acidity (pH < 5) and high salinities (NaCl concentration greater than 10%) [2].

Extremophilic archaea possess unique biochemical and physiological properties that offering vast potential for a wide range of biotechnological applications [3]. Some archaeal products are in developmental stages, such as carotenoids, biohydrogen, polyhydroxyalkanoates and methane [1]. Currently, the only commercially available products from archaea are bacterioruberin, squalene, bacteriorhodopsin and diether/tetraether lipids, which are produced using halophilic archaea [1]. Additionally, archaea represent the most promising sources of extremophilic enzymes [2].

Enzymes from extremophiles, often referred to as extremozymes, are more suitable for industrial applications due to their activity and stability at extreme temperatures and pH ranges. They are adapted to function in high salinity conditions and are tolerant of the presence of metal ions and organic solvents [2,3]. Lipolytic enzymes are hydrolases (E.C. 3.1.1.-) catalyzing the hydrolysis of esters into the corresponding carboxylic acids and alcohols [2,4]. Lipolytic enzymes can be further divided into carboxylesterases (E.C. 3.1.1.1), which specifically hydrolyze water-soluble short acyl chain esters (≤C8), and lipases (EC 3.1.1.3), which specifically hydrolyze water-insoluble long-chain triacylglycerols (≥C8) [2,5,6]. A few archaeal carboxylesterases, belonging to *Pyrobaculum* sp. 1860 [7], *Sulfolobus tokodaii* strain 7 [8], *Nitrososphaera gargensis* [9], *Haloarcula marismortui* [10], *Picrophilus torridus* [11] and *Halobacterium salinarum* NRC-1 [12], have been biochemically characterized [2].

Although archaea have been recognized as a potential source of many extremozymes, including carboxylesterases, the lack of efficient genetic manipulations tools for archaeal biomolecules is a bottleneck for their large-scale production [13]. Expression of heterologous genes in *E. coli* has been a recurrent method to produce different types of foreign proteins. However, many proteins, especially those from halophilic archaea, do not fold properly when expressed heterologous in *E. coli*. Instead, they degrade or accumulate as insoluble aggregates [2,13]. On the other hand, most of the research on homologous expression of halophilic enzymes from archaea has been performed using *Haloferax volcanii* due to its ease of genetic manipulation [14]. *H. volcanii* can be successfully used to overexpress archaeal proteins at medium and large scales [13]. Therefore, it makes sense to use *H. volcanii* for the successful expression of archaeal lipolytic enzymes [2]. In this regard, the aims of this study were: (i) the cloning of a gene of *H. salinarum* NCR-1 encoding for a carboxylesterase in *H. volcanii*, (ii) the induction, purification and immobilization of the recombinant carboxylesterase (rHsEst), and (iii) the biochemical characterization of free and immobilized rHsEst.

## 2. Materials and Methods

### 2.1. Microorganism

*Haloferax volcanii* strain H1209 (ΔpyrE2 ΔhdrB Δmrr Nph-pitA) was generously provided by Dr. Thorsten Allers from the School of Life Sciences, Queens Medical Centre, University of Nottingham.

### 2.2. Culture Media for H. volcanii

The Hv-Ca medium composition was as follows (g L^−1^): NaCl, 144; MgSO_4_•7H_2_O, 21; MgCl_2_•6H_2_O, 18; KCl, 4.2; casamino acids, 5. Additionally, the Hv-Ca medium was supplemented with 12 mL of Tris-HCl (1M, pH 7.5), 5.34 mL of Hv-Ca salts, and 0.66 mL of phosphate buffer (1 M, pH 7.0). The Hv-Ca salts were composed of 6 mL of CaCl_2_ 0.5 M, 1.02 mL of trace elements (containing in mg L^−1^: MnCl_2_•4H_2_O, 0.122; ZnSO_4_•7H_2_O, 0.15; FeSO_4_•7H_2_O, 0.782; CuSO_4_•5H_2_O, 0.017) and 1.32 mL of vitamin solution (containing in mg L^−1^: thiamine, 0.027; and biotin, 0.17) [15].

The Hv-YPC medium composition was as follows (g L^−1^): NaCl, 144; MgSO_4_•7H_2_O, 21; MgCl_2_•6H_2_O, 18; KCl, 4.2; yeast extract, 5; peptone, 1; casamino acids, 1. To one liter of this solution, 12 mL of Tris-HCl (1 M, pH 7.5) and 6 mL of CaCl_2_ (0.5 M), were added [15].

### 2.3. Bioinformatic Analysis

The bioinformatic analysis of the carboxylesterase gene (AAG19778.1, GenBank, Bethesda, MD, USA) from the genome of *H. salinarum* NRC-1 (locus_tag = VNG_RS05745 in entry: NC_002607 from GenomeNet) was conducted using the following tools. Multiple amino acid sequence alignment analyses were performed using MEGA version 11 [16], employing the MUSCLE algorithm. The amino acid sequences were compared with the non-redundant sequence databases deposited at the NCBI (National Center for Biotechnology Information, Bethesda, MD, USA) and the PRALINE Multiple Sequence Alignment [17]. Pairwise structure alignment was conducted with ESPript 3.0 server [18]. Modelling was performed using software programs from the Phyre2 server [19] and AlphaFold2 CoLab [20]. Visualization was conducted with the PyMOL Molecular Graphics System, Version 2.0 Schrödinger, LLC., New York, NY, USA.

### 2.4. Cloning, Overexpression and Purification of Recombinant Carboxylesterase

The plasmid pTA1392, generously provided by Dr. Thorsten Allers, was utilized to clone the carboxylesterase gene the from genome of *H. salinarum* NRC-1. Briefly, the gene AAG19778.1 was synthetized by Epoch Life (Road Missouri City, TX, USA) and cloned into the pUC57-Kan vector. Restriction sites *Pci*I and *Eco*RI were introduced via PCR (FW 5′ CCACATGTTGGA-GACGCTTGCCCAC 3′ and RV 5′ CCACCGGAATTCCCTATCGGCG 3′). The purified PCR product and pTA1392 vector were digested with *Pci*I and *Eco*RI, then purified from agarose gel, ligated with T4 ligase (New England Biolabs, Boston, MA, USA), and subcloned in *E. coli* (DH5α) by electroporation to produce a significant quantity of the plasmid containing the target gene. The final construction (pTA1392-HsEst), resulting in the addition of a 6xHis-tag at the N-terminal of the protein, was used to transform the *H. volcanii* H1209 strain as described by Allers et al. [15].

The transformed *H. volcanii* strain was cultured in Hv-Ca liquid medium at 43 °C and 170 rpm for 72 h. Afterward, 50 mL of this culture was centrifuged at 4500× *g* rpm and 4 °C for 40 min. 25 mL of the supernatant was discarded, and the pellet was resuspended and used to conserve the strain at 4 °C. This concentrate culture was used to inoculate Hv-YPC liquid medium at a 1:10 ratio (to obtain an initial OD_600nm_ of 0.1) and incubated at 43 °C and 170 rpm. Once the culture reached an OD_600nm_ of 0.4, the recombinant carboxylesterase inducer (L-tryptophan) was added to reach a final concentration of 6 mM. The culture was incubated at the above conditions, halted at 16 h, and centrifuged at 4500× *g* rpm and 4 °C for 40 min, recovering the pellet [12,15].

To induce cell disruption, 1 g of fresh biomass was suspended in 20 mL of Tris HCl buffer (50 mM, pH 7.5) at 4 °C and vigorously vortexed twice for 2.5 min [21]. The suspension was then centrifuged at 1792× *g* and 4 °C for 40 min, recovering the supernatant. NaCl and imidazole were added to reach concentrations of 2 M and 10 mM, respectively. A crude extract (20 mL) was filtered through a 0.45 μm membrane and injected into a 1 mL HisTrap HP column to purify the recombinant carboxylesterase (rHsEst) via affinity chromatography. The adsorbed rHsEst was recovered by means of an imidazole concentration gradient (10, 90, 150, 300 and 500 mM). The fractions containing the rHsEst (1.5 mL at 150 mM and 1.5 mL at 300 mM imidazole) were pooled and washed three times with 12 mL of Tris HCl buffer (50 mM, pH 7.5, containing 2 M NaCl), using Amicon® Ultra-15 centrifugal filter units (10 kDa molecular weight cutoff filters) at 3000× *g*, 4 °C for 20 min to remove the imidazole and concentrate the rHsEst.

The purity of rHsEst was verified using electrophoresis (12% SDS-PAGE), and the esterase activity of the band was assessed using a zymogram with 4-methylumberiferyl-butyrate as substrate under non-denaturing conditions. Additionally, the rHsEst was sent to the Instituto de Ecología A. C. (INECOL, Veracruz, Mexico) for sequencing to determine its similarity with the sequence of the protein encoded by the gene AAG19778.1. rHsEst was sequenced by nano LC-MS/MS analysis using an Orbitrap Fusion Tribid mass spectrometer interfaced with an UltiMate 3000 RSLC system (Dionex; Sunnyvale, CA, USA) and set with an ‘EASY Spray’ nano ion source (Thermo-Fisher Scientific, San Jose, CA, USA) [22].

### 2.5. Standard Assay of Esterase Activity

The esterase activity of (free) rHsEst was determined spectrophotometrically by measuring the rate of hydrolysis of *p*-nitrophenyl valerate (p-NPC5). The substrate solution was prepared by mixing one part of p-NPC5 (10 mM) dissolved in 2-propanol and nine parts of a Tris-HCl buffer (100 mM, pH 8.0, containing NaCl 1 M and Triton X-100 at 2 g L^−1^). The enzyme reaction was carried out in glass tubes by mixing one part of the appropriately diluted enzyme with nine parts of the substrate solution. Tubes were incubated at 30 °C for 10 min, and the release of *p*-nitrophenol (p-NP) was measured in a spectrophotometer at OD_410nm_ (DR/2010, HACH, COLO, Loveland, CO, USA). Assays were performed in triplicate. The absorbance data obtained were compared to a standard curve previously prepared with *p*-NP at concentrations ranging from 10 to 100 μM. One unit of enzyme activity was defined as the µmoles of *p*-NP released per minute of reaction, at 30 °C and pH 8.0 [12]. Protein concentration was measured by the Bradford method [23].

The esterase activity of immobilized rHsEst was determined as described above, with the following modifications: in a glass tube, 20 mg of dry immobilized rHsEst was suspended in 100 µL of Tris-HCl buffer (100 mM, pH 8.0, containing NaCl 1 M), and 900 µL of substrate solution was added and mixed. The tubes were incubated at 30 °C and 170 rpm for 10 min. Assays were performed in triplicate.

### 2.6. Kinetic Parameters of rHsEst

Enzyme kinetic assays were carried out using *p*-nitrophenyl acetate (p-NPC2), *p*-nitrophenyl valerate (p-NPC5), and *p*-nitrophenyl caprylate (p-NPC8) as substrates, the initial concentrations were: 0.02, 0.04, 0.06, 0.08, 1.0, 1.5, 2.0 and 3.0 mM. Initial reaction rates (V_0_) were calculated from the lineal portion of the kinetics of the released product (*p*-nitrophenol) during the hydrolysis reaction at 25 °C. The kinetics for each substrate concentration were performed in triplicate, using Tris-HCl buffer (50 mM, pH 8.0, containing 1 M NaCl).

Kinetic parameters (K_M_ and V_MAX_) were determined by fitting the Michaelis–Menten model to experimental values of V_0_ as a function of substrate concentration, using nonlinear regression [24] and SigmaPlot software (version 12.5 for Windows, Systat Software Inc., Palo Alto, CA, USA).

### 2.7. Effect of Metal Ions, Organic Solvents and Detergents, on the Esterase Activity of rHsEst

Purified rHsEst was exposed to various chemical species under the following experimental conditions: NaCl concentration of 1 M, rHsEst esterase activity of 1.99 ± 0.04 U mL^−1^, rHsEst concentration of 0.37 ± 0.06 mg L^−1^, and incubation at 30 °C for 1 h. At the end of the incubation period, in contact with the different chemical species, the residual esterase activity was subsequently determined using the standard assay. For each experiment, control samples were run simultaneously without the addition of any chemical species that could potentially modify the enzyme activity.

The effect of metal ions on esterase activity was studied by incubating rHsEst with Fe^2+^, Mg^2+^, Ca^2+^, Co^2+^, Mn^2+^, Ba^2+^, K^+^, Hg^+^ and Cu^+^, at two concentrations (1 and 5 mM).

The effect of organic solvents on esterase activity was studied by incubating rHsEst with dimethyl sulfoxide, diethyl ether, benzene, toluene, *n*-hexane and *n*-heptane, at three concentrations (30, 50 and 70% *v*/*v*).

The effect of detergents on esterase activity was studied by incubating rHsEst with Triton X-100, Tween 20, Tween 80, N-lauroylsarcosine, and sodium dodecyl sulfate, at two concentrations (0.1 and 1.0%, *w*/*v*).

### 2.8. Immobilization of rHsEst

Immobilization tests of rHsEst were conducted using commercial supports: Celite 545, Immobead 150P and Lewatit VP OC1600. A rHsEst solution was prepared at 0.226 mg mL^−1^ in Tris-HCl buffer (50 mM, pH 7.5, containing NaCl 2 M). 70 mg of each support and 500 μL of rHsEst solution were placed and mixed into microtubes in triplicate. The microtubes were incubated at 25 °C and 150 rpm, for 24 h. Free (non-adsorbed) esterase activity and protein concentration were determined in the aqueous phases. Then, the aqueous phases were discarded, and the supports were washed three times with 1.6 mL Tris-HCl buffer (50 mM, pH 7.5) and dried in an incubator at 30 °C for 48 h. Finally, the immobilization yield and effective immobilization of rHsEst on the different supports were calculated as follows:(1)$$Immobilization\;yield\;\left(\%\right)=\frac{rHsEst{\;}_{0}-rHsEst_{S}\;}{rHsEst_{0}}\left(100\right).$$
where *rHsEst*_0_ represents the initial protein concentration (mg mL^−1^) and *rHsEst_S_* represents the concentration in the supernatant after immobilization (mg mL^−1^).
(2)$$Effective\;immobilization\;\left(\%\right)=\frac{\mathrm{Activity}\;\mathrm{of}\;\mathrm{immobilized}\;rHsEst\;}{\mathrm{Total}\;\mathrm{activity}\;\mathrm{of}\;\mathrm{free}\;rHsEst\;}\left(100\right).$$

### 2.9. Substrate Specificity of Free and Immobilized rHsEst

Substrate specificity was determined using various *p*-nitrophenyl esters: *p*-nitrophenyl acetate, *p*-nitrophenyl propionate, *p*-nitrophenyl butyrate, *p*-nitrophenyl valerate, *p*-nitrophenyl caprylate, *p*-nitrophenyl caprate and *p*-nitrophenyl laurate. Solutions of these substrates were prepared at 10 mM in 2-propanol.

The experimental procedure was the same as that described in the standard assay of esterase activity for both free and immobilized rHsEst. The hydrolysis reaction of *p*-nitrophenyl esters is as follows:
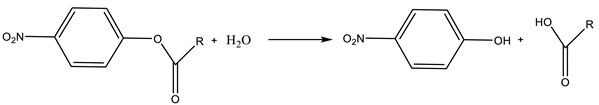
where R corresponds to the fatty acid group (acetate, C2; propionate, C3; butyrate, C4; valerate, C5; caprylate, C8; caprate, C10; and laurate, C12).

### 2.10. Effect of Temperature, pH, NaCl Concentration and Inhibitors on the Esterase Activity of Free and Immobilized rHsEst

The effect of temperature on the esterase activity of free and immobilized rHsEst was studied by incubating at different temperatures (ranging from 20 to 70 °C) for 10 min.

Thermostability of free and immobilized rHsEst was studied by incubating at different temperatures (30, 40, 50 and 60 °C) for 180 h. Experimental data for each incubation temperature were used to calculate the deactivation constants and half-life (*t*_1/2_) using an exponential decay model.

The effect of pH on esterase activity of free and immobilized rHsEst was studied by incubating at 30 °C and different pH levels (ranging from 6 to 10, with 0.5 intervals) for 1 h.

The effect of NaCl concentration on the esterase activity of free and immobilized rHsEst was studied by incubating at 30 °C and different NaCl concentrations (ranging from 0 to 5 M, with 0.5 intervals) for 1 h.

The effect of inhibitors on esterase activity of rHsEst was studied by incubating free and immobilized rHsEst with ethylenediaminetetraacetic acid (EDTA), β-mercaptoethanol and phenylmethylsulphonyl fluoride (PMSF) at 5 mM.

Residual esterase activity was determined using the standard assay after incubation under various experimental conditions.

### 2.11. Statistical Analysis

Analysis of variance (ANOVA) and Tukey’s post hoc tests were performed with a significance level of α = 0.05. ANOVA and multiple comparisons versus control group (Dunnett’s method) were conducted for studies with a significance level of α = 0.05. Statistical analysis was performed using SigmaPlot software version 12.5 for Windows (Systat Software Inc., Palo Alto, CA, USA).

## 3. Results and Discussion

### 3.1. Selection of HsEst gene

Experiments conducted with *H. salinarum* NRC-1 through liquid [12] and solid-state [25] fermentations demonstrated the production of carboxylesterases. However, the enzyme activity achieved in such cultures was so low that its biochemical characterization could not be attained. To overcome this limitation, the carboxylesterase gene (AAG19778.1, GenBank) from the genome of *H. salinarum* NRC-1 (locus_tag = VNG_RS05745 in entry: NC_002607 from GenomeNet) was cloned for expression in *H. volcanii*. Protein-protein blast analysis of HsEst (AAG19778.1) in NCBI server [26] allowed the identification of sequences with high identity with in the *Halobacterium* genus; however, among the sequences with the highest identity, none had been characterized, making HsEst the first to be cloned and characterized in this work. Alignment performed with PRALINE server [17] employing sequences of characterized enzymes which have X-ray structure (Appendix A) showed that AfEst (5FRD) had the highest sequence identity (31.25%).

The in silico analysis of the HsEst sequence revealed a conserved α/β hydrolase domain that is common to esterases and lipases [27]. Furthermore, HsEst contains two regions that are common to esterases and lipases, namely the G-X-S-X-G pentapeptide (residues from 95 to 99), containing the active site serine, and the oxyanion-hole loop (residues from 29 to 31) (Appendix A). HsEst was also found to be homologous to several esterases and lipases from the Hormone Sensitive Lipase (HSL) family, although some differences were observed in both the pentapeptide and oxyanion regions. In the case of the HsEst pentapeptide region, the second residue (96) is an asparagine, whereas other HSL proteins normally contain an aspartate or glutamate residue in this position (Appendix A). A similar case was found for HmEst, where the second residue of the pentapeptide is a histidine (residue 127). Regarding the oxyanion-hole loop, the sequence HGGG is fully conserved among the other members of the HSL family, except for HmEst with RGGA and HsEst with the sequence HGSG (Appendix A).

### 3.2. Structural Modeling of HsEst

To gain structural insight into HsEst, a pairwise alignment with the structure of AfEst esterase (PDB: 5FRD) was performed using ESPript 3.0, revealing the predicted secondary structure and conserved regions (Appendix A). As expected, given the low identity score, only a few regions are conserved, with the most significant being the catalytic site. AfEst is a thermostable esterase (with an optimum temperature at 80 °C) that hydrolyzes monoacylesters of different acyl-chain length and displays maximal activity with *p*-nitrophenyl-esters from C3 to C8 [28], while (free) rHsEst is a mesophilic enzyme, displaying maximal activity with *p*-nitrophenyl-valerate (this work).

The HsEst model structure, obtained with the Phyre2 web [19], was evaluated with MolProbity, obtaining a score of 89.77% for the Ramachandran plot of the PDB file (Appendix A) where five residues (Glu3, Asp75, Leu132, Arg133 and Asp161) were in an unfavorable region of the plot.

The model obtained for HsEst with Phyre2 consists of nine α-helices and a central β-sheet core, containing eight β-strands (Figure 1a). The residues of rHsEst forming the catalytic triad (Ser97, Asp211 and His239), superimpose well onto AfEst residues (Ser89, Asp200 and His228) (Figure 1b,c), despite sharing only 31.25% of sequence identity. Additionally, the model obtained with AlphaFold2 CoLab (AF2C) was similar to the previous model of Phyre2, confirming that the overall fold was as expected. Despite having high confidence in the HsEst models, those should be considered with caution.

The content of negatively and positively charged residues of HsEst and AfEst was obtained with ProtParam tool [29]. For HsEst the Asp + Glu content is 14.7% while the Arg+Lys is 7.6% with a ratio (Asp+Glu)/(Arg+Lys) of 1.93, which is similar to other esterases with major content of negatively charged residues such as Hm EST with a ratio of 2.89 and EST2 with a ratio of 1.54 [29]. Visual charged surface of HsEst model is presented in Appendix A. In the case of AfEst (PDB: 5FRD) the Asp+Glu content is 14.9% while the Arg+Lys is 13% with a ratio (Asp+Glu)/(Arg+Lys) of 1.15 which indicates balance of ionic charges.

### 3.3. Cloning, Overexpression and Purification of Recombinant Carboxylesterase

The AAG19778.1 gene (GeneBank) from the *H. salinarum* NRC-1 genome was identified as encoding the synthesis of a putative carboxylesterase and was selected for cloning. Due to its similar background, *H. volcanii* H1209 was chosen as the ideal host for the recombinant production of the *H. salinarum* NRC-1 carboxylesterase (HsEst). Subsequently, the gene was synthetized and inserted into the pTA1392 vector (Appendix A). The resulting plasmid (pTA1392-HsEst) was cloned into *H. volcanii* H1209 to achieve the inducible expression of the putative enzyme, which contained the addition of a 6x His-tag in the N-terminus to facilitate its purification by affinity chromatography.

*H. volcanii* cells were transformed with pTA1392-HsEst, following the method described by Allers et al. [15], and subsequently cultured in Hv-Ca medium. The clone exhibiting optimal growth in this culture medium was chosen for further experiments. The transformed *H. volcanii* clone was hen cultivated using Hv-YPC medium, and protein expression was induced by supplementing with tryptophan [15]. The production of wet biomass of the transformed *H. volcanii* was 10 g L^−1^, with rHsEst production reaching 14 mg L^−1^ of culture.

The most effective technique for cell lysis was the simplest, most effective technique for cell lysis was the simplest, involving an osmotic shock, achieved by placing the cells in a buffered solution devoid of NaCl. The calculated production of recombinant carboxylesterase from *H. salinarum* NRC-1 (rHsEst) was 70 U L^−1^ of culture medium. Subsequently, rHsEst was purified via affinity chromatography, resulting in a yield of 80.5% and a purification factor of 11.9 (Table 1). Following purification, the protein concentration was 0.77 mg mL^−1^ with an esterase activity of 3.82 U mL^−1^, using *p*-nitrophenyl valerate as the substrate. These purification yields were consistent with values reported by other research groups for recombinant esterases [30,31,32].

The purity of rHsEst was confirmed through electrophoresis (SDS-PAGE), and its esterase activity was validated by band fluorescence resulting from the hydrolysis of 4-methyl-umberiferyl-butyrate, used as a substrate, in a zymogram performed under non-denaturing conditions (Figure 2). The estimated molecular mass of rHsEst by SDS-PAGE was approximately 33.0 kDa, slightly higher than the theoretical molecular mass (~28.8 kDa). This discrepancy could be attributed to the high ratio of negatively/positively charged amino acid residues characteristic of archaeal proteins, leading to aberrant migrations on SDS-PAGE [33]. Indeed, the calculated electrical charges ratio of rHsEst is 1.95 [34]. In comparison, the recombinant esterase LipC from *H. marismortui* expressed in *E. coli* has a negatively/positively charged amino acid residues ratio of 2.89, with an estimated molecular mass in SDS-PAGE of 50 kDa, whereas the theoretical molecular mass is 34 kDa [33]. Conversely, lipase B from *Candida antarctica* (CALB) has a neutral electric ratio of 1.05; therefore, migrations on SDS-PAGE of theoretical and produced recombinant CALB were similar, with minor differences, attributed to post-translational modification (i.e., glycosylation), which depend on the host used for production. As *H. volcanii* has been reported to undergo post-translational modifications, such as *N*-glycosylation [35], an in silico analysis of rHsEst amino acid sequence was conducted, using GlycoPP [36], revealing no putative *N*-glycosylation sites.

The molecular mass of rHsEst was comparable to other reported esterases, such as Est882 from an unknown organism (~32 kDa) [37], Est19 from *Pseudomonas* sp. E2-15 (~35 kDa) [38], esterase from *Bacillus thuringiensis* (~29 kDa) [31], EstGS1 from *Glycomyces salinus* (~25.1 kDa) [39] and CpEST from *Komogataella phaffii* (~38 kDa) [40].

Additionally, the purified rHsEst was sequenced and compared to the theoretical protein sequence of *H. salinarum* NRC-1 (AAG19778.1, GeneBank). Its sequence similarity was confirmed with high confidence (99% confidence and 50% coverage).

### 3.4. Kinetic Parameters of rHsEst Calculated from the Hydrolysis of Three Substrates

The kinetic study of the hydrolysis of three substrates (*p*-nitrophenyl acetate -pNPC2-, *p*-nitrophenyl valerate -pNPC5- and *p*-nitrophenyl octanoate -pNPC8-), catalyzed by rHsEst, was conducted to estimate the enzyme’s kinetic parameters (Table 2). Initial rates (V_0_) were calculated from the kinetics of substrate hydrolysis, whit concentrations [S] ranging from 0.02 to 3.00 mM. Nonlinear regression was used to fit experimental data (V_0_ vs [S]) to the Michaelis-Menten equation (Appendix A) [24]. The best rHsEst kinetic parameters were obtained when using p-NPC5 as a substrate (V_MAX_ = 5.55 μM p-NP s^−1^, K_M_ = 78 µM, k_cat_ = 0.67 s^−1^ and k_cat_/K_M_ = 8.52 s^−1^ mM^−1^). This K_M_ was lower than those reported to EstGS1 from *G. salinus* [39], Est882 (from an unknown organism) [37] and Est1260 (from an unknown organism) [41]; k_cat_ was lower than that reported for Est19 from *Pseudomonas* sp. E2-15 [38]; and k_cat_/K_M_ was lower than those reported for EstGS1 [39], Est882 [37] and Est19 [38]. Remarkably, k_cat_/K_M_ values revealed a low catalytic efficiency of rHsEst, suggesting that rHsEst does not have adequate specificity to catalyze the hydrolysis of these non-natural substrates (*p*-nitrophenyl esters).

### 3.5. Effect of Metal Ions, Organic Solvents and Detergents on the Esterase Activity of rHsEst

Several chemical species could be present in industrial biocatalysis processes. These species may selectively activate or inhibit enzymes by interacting with specific amino acid residues, thereby influencing the enzyme’s catalytic activity [30]. Thus, it is essential to investigate the effect of various chemical species (metal ions, organic solvents, detergents and inhibitors) on the esterase activity of rHsEst. For this study, the following experimental conditions were established: NaCl 1 M, rHsEst esterase activity and protein concentration of 1.99 ± 0.04 U L^−1^ and 0.37 ± 0.06 g L^−1^, respectively, with incubation at 30 °C for 1 h. After the incubation period in contact with different chemical species, the residual esterase activity was determined. Controls for each experimental condition were run simultaneously, without the addition of any chemical species that could modify enzymatic activity.

#### 3.5.1. Effect of Metal Ions on the Esterase Activity of rHsEst

The catalytic activity of rHsEst towards various metal ions was investigated by incubating the purified enzyme with different metal ions (Fe^2+^, Mg^2+^, Ca^2+^, Co^2+^, Mn^2+^, Ba^2+^, K^+^, Hg^+^ and Cu^+^) at concentrations of 1 and 5 mM. rHsEst maintained its activity in the presence of the Mg^2+^, Ca^2+^, Mn^2+^, Ba^2+^ and K^+^ at both concentrations tested (Figure 3a). These results were similar to that reported for EstGS1 from *G. salinus* [39], a *B. thuringiensis* esterase [31] and Est19 from *Pseudomonas* sp. E2-15 [38].

At 1mM, Fe^2+^ did not significantly affect rHsEst; but at 5 mM, it resulted in approximately 30% residual activity compared to the control. These inhibition pattern mirrored that observed for EstGS1 from *G. salinus* [39], but differed from that of Est19 from *Pseudomonas* sp. E2-15 [38]. In the presence of Co^2+^, rHsEst exhibited around 55% residual activity at both concentrations tested, consistent with observations for EstGS1 from *G. salinus* [39], a *B. thuringiensis* esterase [31] and Est19 from *Pseudomonas* sp. E2-15 [38].

Conversely, Hg^+^ and Cu^+^ almost completely inhibited rHsEst activity at both concentrations. This inhibition pattern resembled that reported for EstGS1 from *G. salinus* [39]. Overall, metal ions can interact with amino acid side chains radicals, impacting residue ionization and causing enzyme instability [5,31]. This study demonstrated that the esterase activity of rHsEst remained stable in the presence of most of the metal ions tested. Specifically, rHsEst exhibited stability in the presence of the most common redox-inert metal ions (Mg^2+^ and Ca^2+^) as well as two redox-active metal ions (Mn^2+^ and Co^2+^) [42].

#### 3.5.2. Effect of Solvents on the Esterase Activity of rHsEst

The tolerance of enzymes to solvents is a fundamental requirement in many industrial biosynthesis processes [30]. Hence, it was important to study the effect of organic solvents on the stability of rHsEst. The explored solvents were dimethyl sulfoxide, diethyl ether, benzene, toluene, *n*-hexane and *n*-heptane at three concentrations: 30, 50 and 70%, *v*/*v* (Figure 3b).

In the presence of DMSO, the residual esterase activity of rHsEst was 90%, 18% and 1% for solvent concentrations of 30%, 50% and 70%, respectively. These results were similar to those reported for EstD04 from *Pseudomonas* sp. D01 [5], EstGS1 from *G. salinus* [39], EstD9 from *Anoxybacillus geothermalis* D9 [30] and Est19 from *Pseudomonas* sp. E2-15 [38]. In the presence of diethyl ether, the residual esterase activity of rHsEst was around 112% at 30% and 50% of the solvent and decreased to 78% at 70% of the solvent.

In the presence of benzene, the residual esterase activity of rHsEst was around 47% at 30% and 50% of the solvent and decreased to 24% at 70% of the solvent. In the presence of toluene, the residual esterase activity of rHsEst was, on average, 61% at 30% and 50% of the solvent and decreased to 27% at 70% of the solvent. These results, with benzene and toluene, were similar to those reported for EstD9 from *A. geothermalis* D9 [30].

In the presence of *n*-hexane, the residual esterase activity of rHsEst was around 82 at 30% and 50% of the solvent and decreased to 67% at 70% of the solvent. These results were similar to those reported for EstD04 *Pseudomonas* sp. D01 [5], while rHsEst had greater tolerance to *n*-hexane than that reported for EstDZ3 from *Dictyoglomus* sp. [43]. In the presence of *n*-heptane, the residual esterase activity of rHsEst was 86% at 30% of the solvent and decreased around to 66%, at 50% and 70% of the solvent. These results were similar to those reported for Est19 from *Pseudomonas* sp. E2-15 [38].

Overall, rHsEst showed great stability in the presence of diethyl ether, *n*-hexane, and *n*-heptane; therefore, rHsEst could be applied in organic synthesis using these solvents.

#### 3.5.3. Effect of Detergents on the Esterase Activity of rHsEst

A more comprehensive understanding of the operational stability of enzymes in the presence of detergents is essential for potential industrial applications [30]. Therefore, the stability of rHsEst in the presence of detergents (Triton X-100, Tween 20, Tween 80, N-Lauroyl sarcosine and sodium dodecyl sulfate) at two concentrations (0.1 and 1.0%, *w*/*v*) was studied (Figure 3c).

rHsEst maintained its activity completely at 0.1% of Triton X-100, Tween 20, and Tween 80. However, at 1.0% of these detergents, the residual esterase activity of rHsEst decreased to an average of 23% for Triton X-100 and Tween 20 and to 12% for Tween 80. In the presence of NLS and SDS, the residual esterase activity of rHsEst was around 17% at both concentrations tested. These results were similar to those reported for Est19 from *Pseudomonas* sp. E2-15 [38]. However, rHsEst exhibited less tolerance to detergents than that reported for EstD04 from *Pseudomonas* sp. D01 [5], EstGS1 from *G. salinus* [39] and EstD9 from *A. geothermalis* D9 [30]. Overall, rHsEst showed limited stability in the presence of detergents.

### 3.6. Immobilization of rHsEst

Enzyme immobilization is widely used for industrial applications since it reduces costs, allows enzyme recovery, and increases enzyme operational stability and product yields [37,44,45]. Hence, the immobilization of rHsEst was carried out using commercial supports, including Celite 545, Immobead 150P and Lewatit VP OC1600 (Table 3). rHsEst immobilization with Celite 545 and Lewatit VP OC1600 yielded 40% and 42% of non-adsorbed protein (assayed in the aqueous phase), respectively. Mean-while, rHsEst immobilization with Immobead 150P yielded 7% of non-adsorbed protein (assayed in the aqueous phase). However, immobilized rHsEst on Immobead 150P and Lewatit VP OC1600 did not show esterase activity, revealing that those immobilizations were not effective. These results can be attributed to the distinct physicochemical features of each matrix, as well as to pore size and the nature of the reactive groups. The free epoxy groups present in these supports can react with rHsEst, affecting its structural conformation and/or blocking its active site. [46,47]. On the other hand, immobilized rHsEst on Celite 545 yielded an effective immobilization of 47% (0.318 U g^−1^ of support). This immobilization yield was similar to those reported for lipases immobilized in Celite 545 [48,49].

### 3.7. Substrate Specificity of Free and Immobilized rHsEst

The substrate specificity of free rHsEst was determined by hydrolyzing various *p*-nitrophenyl (pNP) esters with variable carbon chain lengths (C2, C3, C4, C5, C8, C10 and C12) (Figure 4a). Free rHsEst specifically hydrolyze short-chain *p*-nitrophenyl esters (C2-C5), exhibiting the high enzymatic activity with *p*-nitrophenyl valerate (pNPC5) with a specific esterase activity of 3.02 ± 0.01 U mg^−1^. This chain length preference for *p*-nitrophenyl esters consistent with previous reports for CpEST from *K. phaffii* (pNPC4) [40], EstD9 from *A. geothermalis* D9 (pNPC2-pNPC4) [30], Est882 (pNPC2) [37] and Est19 from *Pseudomonas* sp. E2-15 (pNPC6) [38].

Furthermore, the substrate specificity of immobilized rHsEst was determined using the above substrates. Figure 4b shows higher esterase activities using pNPC3, pNPC4 and pNPC5, with no significant differences (*p* = 0.05). Relative esterase activity decreased by 50% using pNPC2, pNPC8 and pNPC10, and it decreased by 80%, using the pNPC12. These results indicated that immobilized rHsEst maintains its specificity for hydrolyzing short-chain *p*-nitrophenol esters. However, unlike free rHsEst, immobilized rHsEst was able to hydrolyze pNPC10 and pNPC12, and it exhibited the highest catalytic activity using pNPC3, pNPC4 and pNPC5 as substrates, rather than only pNPC5 for the free enzyme. These results confirm that the recombinant enzyme (rHsEst) is a carboxylesterase [30,38].

### 3.8. Effect of Temperature, pH, NaCl Concentration and Inhibitors on the Esterase Activity of Free and Immobilized rHsEst

Temperature, pH, NaCl concentration and presence of inhibitors are key factors influencing activity and stability enzyme. Therefore, characterizing the esterase activity of rHsEst as a function of incubation temperature, pH and NaCl concentration, as well as its thermostability and tolerance to inhibitors, is of significant interest for potential industrial applications [30,45].

#### 3.8.1. Effect of Incubation Temperature on Esterase Activity and Stability of Free and Immobilized rHsEst

The effect of incubation temperature (ranging from 20 to 70 °C) on the esterase activity of free and immobilized rHsEst was studied using *p*-nitrophenyl valerate as substrate. The optimal temperature for the activity of both free and immobilized rHsEst was determined to be 30 °C (Figure 5). Notably, it was observed that at incubation temperatures exceeding 60 °C, immobilized rHsEst lost 56% of its activity, while free rHsEst experienced a more significant decrease of 86%. These results were similar to those reported for Est882 (from an unknown organism) [37], Est_BAS_ΔSP from *Bacillus altitudinis* [45] and CALB from *C. antarctica* [46].

The thermostability of free and immobilized rHsEst was studied, by incubating them at different temperatures (30, 40, 50 and 60 °C) for 180 min (Figure 6). Free rHsEst was only stable at 30 °C, with a half-life time of 10.2 h. It was moderately inactivated at 40 °C, with a half-life time of 2.73 h, and quickly inactivated at 50 °C and 60 °C, with half-life times of 0.72 and 0.10 h, respectively (Figure 6a). On the other hand, immobilized rHsEst was fully thermostable at 30 and 40 °C, with half-life times of 105 and 10.1 h, respectively (Figure 6b). Additionally, immobilized rHsEst was more stable than free rHsEst at 50 °C and 60 °C, with half-life times of 2.97 and 1.58 h, respectively. These results were similar to those reported for other immobilized esterases [47,50]. This increase in thermostability of immobilized rHsEst can be attributed to the protection provided by the support’s rigid structure [45].

#### 3.8.2. Effect of pH and NaCl Concentration on Esterase Activity of Free and Immobilized rHsEst

The effect of pH (ranging from 6 to 10) on the esterase activity of free and immobilized rHsEst was studied (Figure 7a). Free rHsEst showed optimal activity at pH 8.0, while immobilized rHsEst exhibited its highest activity at pH 10.0. In the pH range of 7.5 to 9.0, free rHsEst retained at least 80% of its activity. However, at pH levels lower than 7.5 and higher than 9.0, free rHsEst lost more than 60% of activity. Similarly, at pH levels below 8.0, immobilized rHsEst also lost more than 60% of its esterase activity. The optimal pH range (8–10) for immobilized rHsEst activity was similar to what has been reported for other esterases [30,37,44,45]. These results revealed that rHsEst immobilization supported a more stable enzyme conformation over a wide pH range [45].

The effect of NaCl concentration (ranging from 0 to 5 M) on the esterase activity of free and immobilized rHsEst was studied, considering that rHsEst is an enzyme derived from a halophilic archaeon (*H. salinarum* NCR-1) (Figure 7b). Free rHsEst showed optimal activity at 1 M NaCl, whereas immobilized rHsEst exhibited a wider optimal activity range (from 0 to 3 M NaCl). At NaCl concentrations of 0, 3 and 3.5 M, more than 50% of free rHsEst relative activity was retained, while at NaCl concentrations of 3.5, 4.0 and 4.5 M, more than 70% of immobilized rHsEst relative activity was maintained. When the NaCl concentrations exceeded 3.5 M, the relative activity of free rHsEst decreased to 19%, while that of immobilized rHsEst only decreased to 38% at 5.0 M NaCl. This salt tolerance of rHsEst was similar to what has been reported for Est1360WT (from an unknown organism) [41] and EstGS1 from *G. salinus* [39]. The increase in rHsEst’s salt tolerance could be explained by a more stable enzyme conformation due to immobilization.

#### 3.8.3. Effect of Inhibitors on Esterase Activity of Free and Immobilized rHsEst

Finally, the stability of immobilized rHsEst in the presence of inhibitors (EDTA, BME and PMSF) at 5 mM was studied (Figure 8). Immobilized rHsEst maintained full activity in the presence of EDTA, while free rHsEst reduced its activity to 53%. This indicates that the fold and/or activity of rHsEst depend on some metal co-factor, contrasting with what has been reported for others esterases [38,39,43]. In the presence of BME, the residual esterase activities of immobilized and free rHsEst showed a slight reduction, to around 78%. This result suggests that the presence of cysteine and disulfide bridges are not important for the activity and stability of rHsEst. In the presence of PMSF, the residual esterase activity of immobilized rHsEst decreased to 71%, while free rHsEst reduced it to 8%. The almost total inhibition of enzymatic activity by PMSF reveals the presence of a serine residue in the active site of rHsEst [30,51]. These results indicated that rHsEst immobilization effectively protects it from the action of the inhibitors tested.

## 4. Conclusions

The gene encoding a halophilic carboxylesterase (AAG19778.1) from *H. salinarum* NRC-1 was successfully cloned and expressed in *H. volcanii,* due to compatibility of codon usage. The overexpression of recombinant esterase (rHsEst) was inducted with tryptophan in a culture medium containing 2.5 M NaCl, followed by purification via affinity chromatography. Its molecular weight was estimated to be 33 kDa by SDS-PAGE. The best kinetic parameters of rHsEst were achieved using *p*-nitrophenyl valerate as substrate (K_M_ of 78 µM and k_cat_ of 0.67 s^−1^). Furthermore, rHsEst demonstrated remarkable stability in the presence of metal ions (Fe^2+^, Mg^2+^, Ca^2+^, Mn^2+^, Ba^2+^ and K^+^) and solvents (diethyl ether, *n*-hexane and *n*-heptane) highlighting its suitability to be used for diverse environmental conditions. Purified rHsEst was effectively immobilized using Celite 545. The carboxylesterase activity of both free and immobilized rHsEst was confirmed through a substrate specificity study. The presence of a serine residue in rHsEst’s active site was revealed by the enzyme activity inhibition by PMSF. The optimal incubation temperature for activity of both free and immobilized rHsEst was 30 °C, while the optimal pH for free rHsEst was 8, and immobilized rHsEst showed optimal activity within a pH range between 8 and 10. Immobilization of rHsEst increased its thermostability, halophilicity and protection against inhibitors as EDTA, BME and PMSF. Remarkably, immobilized rHsEst remained stable and active even at NaCl concentrations as high as 5 M. These findings contribute significantly to the field of biocatalysis and immobilized rHsEst represents a promise for the development of a novel and efficient biocatalyst for potential industrial applications.

## Figures and Tables

**Figure 1 biomolecules-14-00534-f001:**
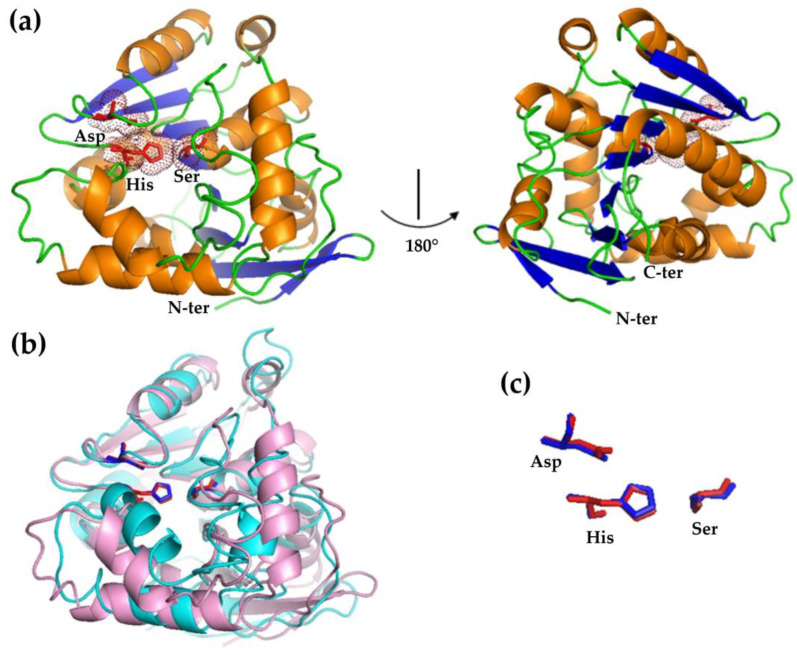
Cartoon representation of the rHsEst model obtained using the Phyre2 server (**a**). α-helices are depicted in orange, β-strands in blue and the active site residues (Ser103, Asp217 and His245) are shown in red. Superimposition between the rHsEST model (magenta) and the AfEst (PDB: 5FRD) crystal structure (cyan) (**b**). Superimposition of the residues of the catalytic triad, depicted in red for rHsEst and blue for EST2 (**c**). Molecular visualization was generated with Pymol.

**Figure 2 biomolecules-14-00534-f002:**
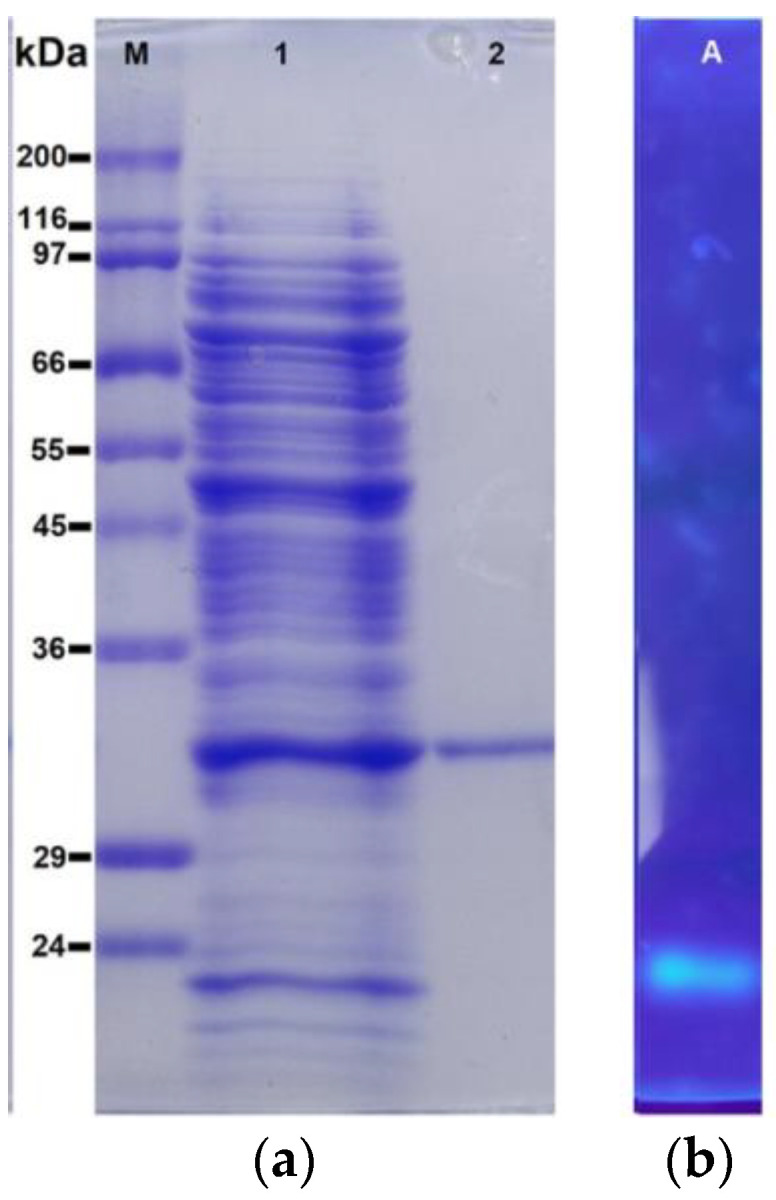
SDS-PAGE analysis of the intracellular crude extract (lane 1) and the purified rHsEst (lane 2) produced by recombinant *H. volcanii*. Lane M: standard molecular weight marker proteins (**a**). Zymogram revealing rHsEst esterase activity through band fluorescence, resulting from the hydrolysis of 4-methylumbelliferyl butyrate used as a substrate under non-denaturing conditions (**b**).

**Figure 3 biomolecules-14-00534-f003:**
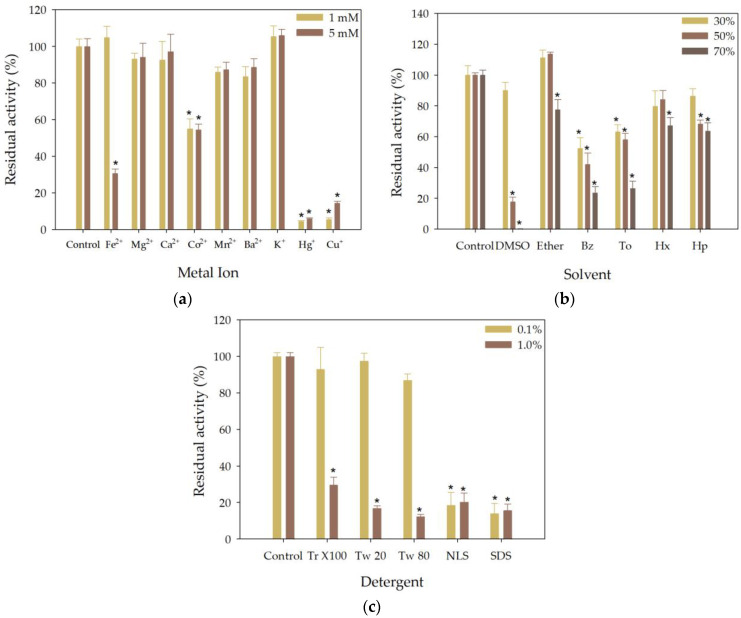
Effect of metal ions (Fe^2+^, Mg^2+^, Ca^2+^, Co^2+^, Mn^2+^, Ba^2+^, K^+^, Hg^+^ and Cu^+^) (**a**), solvents (dimethyl sulfoxide -DMSO-, diethyl ether -Ether-, benzene -Bz-, toluene -To-, *n*-hexane -Hx- and *n*-heptane -H-) (**b**), and detergents (Triton X-100 -Tr X100-, Tween 20 -Tw 20-, Tween 80 -Tw 80-, N-Lauroyl sarcosine -NLS- and sodium dodecyl sulfate -SDS-) (**c**) on the residual esterase activity of rHsEst. The assayed concentrations were: 1 and 5 mM for metal ions; 30%, 50% and 70% (*v*/*v*) for solvents; 0.1 and 1.0% (*w*/*v*) for detergents. Experimental conditions were: NaCl 1 M, rHsEst esterase activity of 1.99 ± 0.04 U L^−1^, rHsEst protein concentration of 0.37 ± 0.06 g L^−1^, and incubation at 30 °C for 1 h. An asterisk indicates significant differences (α = 0.05, Dunnett´s method). Results are the average ± SE of three independent measurements.

**Figure 4 biomolecules-14-00534-f004:**
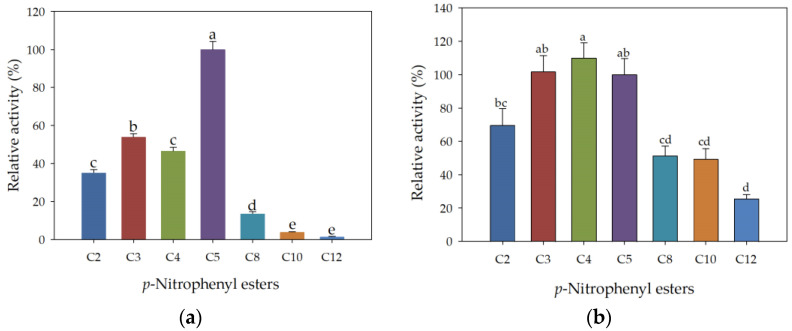
Substrate specificity of free (**a**) and immobilized (**b**) rHsEst using *p*-nitrophenyl (pNP) esters with variable carbon chain lengths (C2, C3, C4, C5, C8, C10 and C12). Esterase activity on pNPC5 was considered as the 100%. Different letters indicate significant differences (α = 0.05, Tukey test). Results represent the average ± SE of three assays.

**Figure 5 biomolecules-14-00534-f005:**
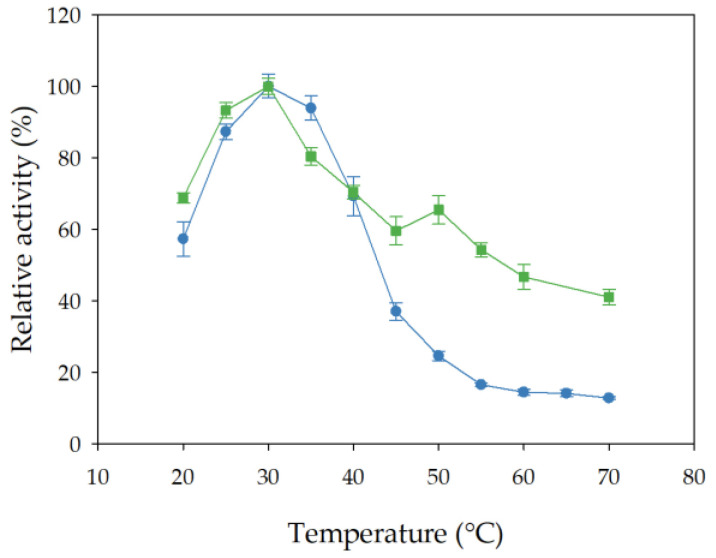
Effect of incubation temperature on esterase activity of free (-●-) and immobilized (-■-) rHsEst. Esterase activity was assayed using *p*-nitrophenyl valerate as substrate. Results represent the average ± SE of three independent measurements.

**Figure 6 biomolecules-14-00534-f006:**
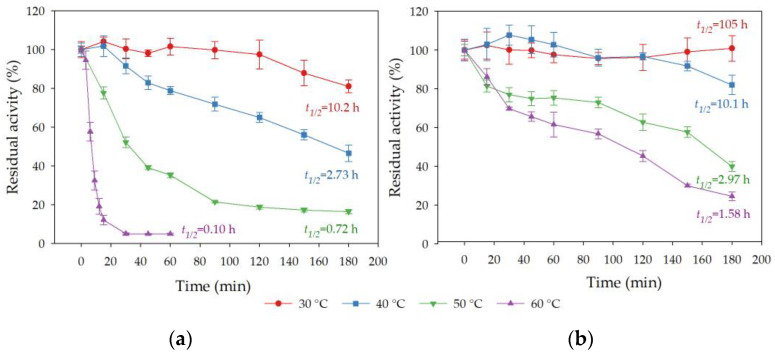
Thermostability of free rHsEst (**a**) and immobilized rHsEst (**b**). Residual esterase activity was assayed using *p*-nitrophenyl valerate as substrate. *t*_1/2_ is the half-life time calculated. Results represent the average ± SE of three independent measurements.

**Figure 7 biomolecules-14-00534-f007:**
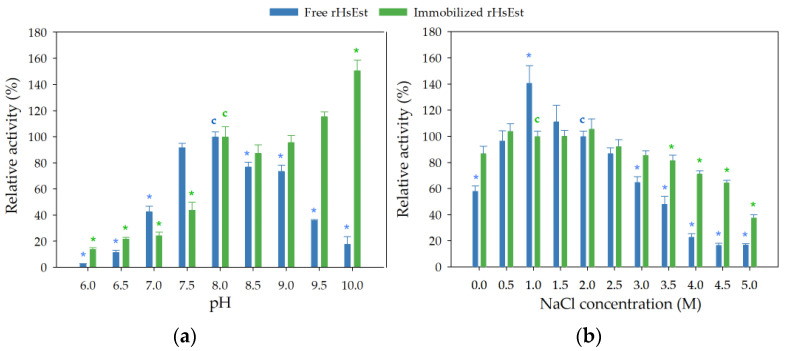
Effect of pH (**a**) and NaCl concentration (**b**) on the activity of free and immobilized rHsEst. Esterase activity was assayed using *p*-nitrophenyl valerate as substrate. Asterisks indicate significant differences (α = 0.05, Dunnett´s method). The treatments used as controls (pH 8, NaCl 1 M and 2 M) were considered as 100% and are marked with the letter “c”. Results represent the average ± SE of three independent measurements.

**Figure 8 biomolecules-14-00534-f008:**
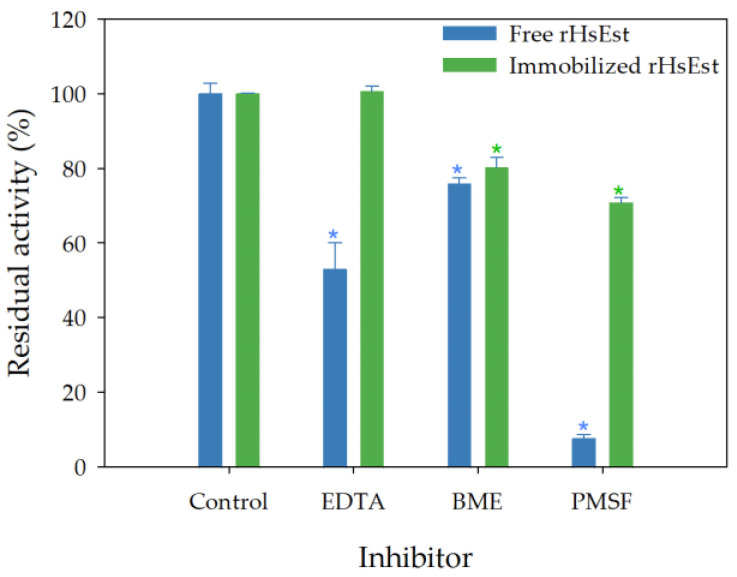
Effect of inhibitors (ethylenediaminetetraacetic acid -EDTA-, β-mercaptoethanol -BME- and phenylmethyl-sulphonyl fluoride -PMSF-) at 5 mM, on esterase activity of free and immobilized rHsEst. Esterase activity was assayed using *p*-nitrophenyl valerate as substrate. Esterase activity of controls (not supplemented with any inhibitor) was considered 100%. Asterisk indicates significant differences (α = 0.05, Dunnett´s method). Results represent the average ± SE of three independent measurements.

**Table 1 biomolecules-14-00534-t001:** Affinity chromatography purification of recombinant carboxylesterase from *H. salinarum* NRC-1 (rHsEst) produced by transformed *H. volcanii*.

Step	Total Activity (U)	Total Protein (mg)	Specific Activity (U mg^−1^)	Yield %	Purification Factor
Crude extract	78.2	185	0.4	100.0	1.00
HisTrap HP column	63.0	13	5.0	80.5	11.90

**Table 2 biomolecules-14-00534-t002:** Kinetic parameters estimated from the Michaelis-Menten model fitted to hydrolysis initial rates (V_0_) as a function of substrate concentrations [S], catalyzed by rHsEst.

Substrate	V_MAX_ (μM pNP s^−1^)	K_M_ (µM)	k_cat_ (s^−1^)	k_cat_/K_M_ (s^−1^ mM^−1^)
pNPC2	0.68 ± 0.08	258 ± 51	0.02	0.07
pNPC5	5.55 ± 0.21	78 ± 7	0.67	8.52
pNPC8	3.33 ± 0.20	245 ± 26	0.09	0.37

Substrate concentrations ranged from 0.02 to 3 mM. Experimental conditions for hydrolysis reactions were 30 °C, pH 8, and NaCl 1 M. V_MAX_ is the maximum rate; K_M_ is the Michaelis-Menten constant; k_cat_ is the turnover number. pNP, pNPC2, pNPC5 and pNPC8 are *p*-nitrophenol, *p*-nitrophenyl acetate, *p*-nitrophenyl valerate and *p*-nitrophenyl octanoate, respectively. Results are expressed as the average ± SE of three independent assays.

**Table 3 biomolecules-14-00534-t003:** Immobilization of rHsEst on different commercial supports.

Supports	Free Protein (mg mL^−1^)	Adsorbed Protein (mg g^−1^ of Support)	Immobilization Yield of Protein %	Free Esterase Activity (U mL^−1^)	Immobilized Esterase Activity (U g^−1^)	Immobilization Yield of Esterase Activity %
Celite 545	0.091 ± 0.013	0.96 ± 0.04	60	0.229 ± 0.004	0.318 ± 0.008	47
Immobead 150P	0.016 ± 0.007	1.50 ± 0.09	93	0.002 ± 0.001	0	0
Lewatit VP OC1600	0.096 ± 0.010	0.93 ± 0.05	58	0.175 ± 0.005	0	0

70 mg of each support and 500 μL of rHsEst solution (at 0.226 mg protein mL^−1^ and 0.671 U mL^−1^) were incubated at 25 °C and 150 rpm for 24 h. Esterase activity was assayed using *p*-nitrophenyl valerate as a substrate at 30 °C and pH 8. Free protein and esterase activity were determined in the aqueous phase. Results are the average ± SE of three independent measurements.

## Data Availability

The data presented in this study are available upon request from the corresponding author. The data are not publicly available due to the large amount of information that needed to be synthesized to be presented in this paper.

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
