# Peer review of "Cloning, Expression, Characterization and Immobilization of a Recombinant Carboxylesterase from the Halophilic Archaeon, Halobacterium salinarum NCR-1"

_biomolecules, 2024, doi:10.3390/biom14050534_

Round 1
Reviewer 1 Report (Previous Reviewer 2)
Comments and Suggestions for Authors
The authors have improve technical aspects of the manuscript. However, the part in which comparison of sequences and structures/models are presented has not improved, but only got more confusing. The authors should clearly explain/present data concerning the sequence similarity with known esterases. It is clear that that are many close homologs, but what is relevant is to show how similar/different the esterase when compared with known esterases, specifically esterase for which structures have been deposited. For example, when doing a BLASTP in the PDB, the PDB 5FRD pops up first. This is not discussed in the paper. Instead, for unknown reasons the PDB 1U4N is used. Why no 5FRD? Also, why include a His-tag in modeling?? Is it expected that that has a particular interaction with the native protein? What is AfEst, why is it included in the analysis?
The inclusion of the AF2-predicted structure is done in a very sloppy manner. The text is confusing. It would be better to base the analysis of the predicted structure, and perhaps just mention that homology modeling gives something similar. Anyway, the whole modeling part is not that important: shorten it and just conclude that the overall fold is as expected (unless you add data on mutants). What are 'punctual mutations'?
How can Vmax values of several substrates vary that much with the kcat values (difference between the values for NPC2 is >10x while for NPC5 it is <10x)?
Comments on the Quality of English LanguageSome parts really need to improve.
Author Response
Dear Reviewer 1: We deeply appreciate your comprehensive and detailed review of our paper. We appreciate your great contribution to have a better publication.
Questions to reply:
- The authors should clearly explain/present data concerning the sequence similarity with known esterases. It is clear that that are many close homologs, but what is relevant is to show how similar/different the esterase when compared with known esterases, specifically esterase for which structures have been deposited. For example, when doing a BLASTP in the PDB, the PDB 5FRD pops up first. This is not discussed in the paper. Instead, for unknown reasons the PDB 1U4N is used. Why no 5FRD? Also, why include a His-tag in modeling?? Is it expected that that has a particular interaction with the native protein? What is AfEst, why is it included in the analysis?
Response:
We agree with your comment. Only characterized esterases with deposited structure should be relevant to compare with our esterase (HsEst). However, it is important to point out that previously (in the first round), other Reviewer suggested the inclusion of close homologues.
The related information to non-characterized homologs was deleted in Section 3.1 (Selection of HsEst gene) and Figure S1 was modified to include only primary structure comparison with known and relevant esterases.
We appreciate your suggestion of using PDB: 5FRD. This crystal structure, which encodes for Archaeoglobus fulgidus esterase (AfEst), is a known and characterized thermostable esterase with a sequence identity of 31.25% with our esterase (HsEst). Consequently, PDB: 5FRD was used as template for structure alignment with the HsEst model (obtained with Phyre2 which is a program based on structural homologies) instead of PDB: 1U4N, which has lower sequence identity. This modification is shown in Figure 1.
In addition, a surface charge of HsEst model is included (Figure S5 in Supplementary Data), discussed and compared to other characterized esterases at the end of section 3.2 (Structural modeling of HsEst), in lines 297-304.
Following your suggestion, the His-tag in N-terminal was not included in the model of HsEst, because we do not expect a particular interaction with the native protein. This modification can be observed in Figure 1 and Figure S3.
- The inclusion of the AF2-predicted structure is done in a very sloppy manner. The text is confusing. It would be better to base the analysis of the predicted structure, and perhaps just mention that homology modeling gives something similar. Anyway, the whole modeling part is not that important: shorten it and just conclude that the overall fold is as expected (unless you add data on mutants). What are 'punctual mutations'?
Response:
We agree with your comments. Since the related information to AF-2 is not relevant, it was deleted in all the figures and text. Only a phrase mentioning the similarity of AF-2 with the model presented in Figure 1, is now included in Section 3.2 (Structural modeling of HsEst). Finally, the 'punctual mutations' phrase was modified.
3) How can Vmax values of several substrates vary that much with the kcat values (difference between the values for NPC2 is >10x while for NPC5 it is <10x)?
Response:
The high differences in Vmax values were due to the enzyme concentrations; which were different for each enzyme kinetic, using a specific substrate. It was a mistake that we made.
The initial velocities were standardized, considering the enzyme concentration used for each substrate and the Vmax values were recalculated in Table 2. The new Vmax values were congruent. Figure S7 was also updated with the new values.
Comments on the Quality of English Language: Some parts really need to improve.
Response:
The whole document was reviewed and corrected by two colleagues with a high level of English.
Reviewer 2 Report (New Reviewer)
Comments and Suggestions for Authors
I have carefully reviewed the manuscript, which presents the biochemical characterization of a recombinant carboxylesterase (rHsEst) derived from Halobacterium salinarum NRC-1. The study successfully cloned and expressed the gene encoding carboxylesterase, followed by purification and characterization of the enzyme. Various enzymatic properties, substrate specificity, stability under different conditions, and the effects of immobilization on enzyme properties were investigated. The findings suggest that immobilized rHsEst exhibits promising biochemical characteristics, indicating its potential for industrial applications. Overall, the manuscript is well-organized and interesting, and it may merit publication in Biomolecules after addressing the following comments.
Comments:
- The supplementary data should be in a separate file. Currently, they are included after the inclusion section. Please remove them and add them to the Word file, including the manuscript information, supplementary data, and their captions.
- It is recommended to transfer Fig. 2 to the supplementary data.
- The authors reported that the immobilization of the esterase enzyme on immobead and Lewatit supports showed 0% expressed activity. Please provide specific reasons for this observation in the discussion section. Potential factors such as surface properties, pore size, or compatibility issues should be considered.
- For a perfect comparison in Figure 5, please display the enzyme activity as "U/mg" for both free and immobilized enzyme forms.
- The reusability of a biocatalyst is a crucial feature. Therefore, please include the reusability results for immobilized esterase in the manuscript. Provide details on the number of cycles performed and any observed changes in enzyme activity over time.
- Please include the storage stabilities for both free and immobilized esterase enzymes. This information is important for assessing the practical applicability and shelf-life of the enzyme preparations.
Minor editing of English language required
Author Response
Dear Reviewer 2: We deeply appreciate your comprehensive and detailed review of our paper. We appreciate your great contribution to have a better publication.
Questions to reply:
- The supplementary data should be in a separate file. Currently, they are included after the inclusion section. Please remove them and add them to the Word file, including the manuscript information, supplementary data, and their captions.
Response:
The supplementary data were separated in a new Word file.
- It is recommended to transfer Fig. 2 to the supplementary data.
Response:
Figure 2 was transferred to the supplementary data and renamed Figure S6.
- The authors reported that the immobilization of the esterase enzyme on immobead and Lewatit supports showed 0% expressed activity. Please provide specific reasons for this observation in the discussion section. Potential factors such as surface properties, pore size, or compatibility issues should be considered.
Response:
Immobilized rHsEst on Immobead 150P and Lewatit VP OC1600 did not show esterase activity. These results can be attributed to the distinct physicochemical features of each matrix, as well as to pore size and the nature of the reactive groups. Particularly, the free epoxy groups present in these supports, may react with the enzyme, affecting its structural conformation and/or blocking its active site. This discussion was added to the document (lines 484-486).
- For a perfect comparison in Figure 5, please display the enzyme activity as "U/mg" for both free and immobilized enzyme forms.
Response:
The objective of Section 3.7 was to determine the substrate specificity for each form of rHsEst (free and immobilized). For a better comparison between these forms, more studies are necessary on the immobilized rHsEst. The amount of protein that is immobilized is known, but it is unknown if all the immobilized protein maintains its esterase activity. For this reason, “U/mg” was not used to compare between free and immobilized enzyme forms.
- The reusability of a biocatalyst is a crucial feature. Therefore, please include the reusability results for immobilized esterase in the manuscript. Provide details on the number of cycles performed and any observed changes in enzyme activity over time.
Response:
This work was focused primarily on the molecular characterization of the esterase of an extremophilic archaeon (H. salinarum, an extreme halophilic). Your comment regarding the reusability of the biocatalyst is very relevant and exciting; However, if you have no objection, we would like to reserve it for a future publication.
In contrast, Figure 7 shows the stability of the free and immobilized esterases, at different incubation temperatures. This figure shows the full stability of the immobilized enzyme, when it is incubated at 30°C (which is the optimal temperature for enzymatic activity).
- Please include the storage stabilities for both free and immobilized esterase enzymes. This information is important for assessing the practical applicability and shelf-life of the enzyme preparations.
Response:
The storage stabilities (for long term) for both free and immobilized esterases are also considered for a future publication. This experiment is currently in progress.
The whole document was reviewed and corrected by two colleagues with a high level of English.
Round 2
Reviewer 2 Report (New Reviewer)
Comments and Suggestions for Authors
The responses and the changes made to the revised manuscript are appropriate and I recommend that this paper be accepted for publication in Biomolecules journal.
This manuscript is a resubmission of an earlier submission. The following is a list of the peer review reports and author responses from that submission.
Round 1
Reviewer 1 Report
Comments and Suggestions for Authors
The authors report the cloning, expression, characterization and immobilization of the recombinant carboxylesterase from Halobacterium salinarum NCR-1. The study is quite comprehensive. To improve the value of this paper, please consider the following comments.
1) Page 9, section 3.4.1 Please evaluate effect of the type of metal ions on enzyme activity.
2) Page 10, section 3.4.3 Detergents at low concentrations were investigated, how about their impacts at high concentrations?
3) Page 12, Figure 5 Please provide chemical reaction formulas of various substrates catalyzed by the enzyme.
Comments on the Quality of English LanguageThe Quality of English Language is good. Minor revision is needed.
Author Response
Dear Reviewer 1: We deeply appreciate your comprehensive and detailed review of our paper. We appreciate your great contribution to have a better publication.
Questions to reply:
1) Page 9, section 3.4.1 Please evaluate effect of the type of metal ions on enzyme activity.
Response: Overall, metal ions can interact with the radicals of amino acid side chains, considerably affecting the ionization of the aminoacid residues and, thus, causing the instability of enzyme. In particular, rHsEst was stable in the presence of most common redox-inert metal ions (Mg2+ and Ca2+) and two redox-active metal ions (Mn2+ and Co2+). These comments were added in lines 419 and 420.
2) Page 10, section 3.4.3 Detergents at low concentrations were investigated, how about their impacts at high concentrations?
Response: No, we did not teste higher concentrations. Detergent concentrations used in our work, were inspired by other reports, where the biochemical characterization of lipolytic enzymes was addressed. At the higher detergent concentration tested in our study, most of lipolytic enzymes completely lose their activity.
3) Page 12, Figure 5 Please provide chemical reaction formulas of various substrates catalyzed by the enzyme.
Response: The general formula for the chemical reaction and the substrates description are provided in section 2.9, lines 206 -216.
Reviewer 2 Report
Comments and Suggestions for Authors
The authors report on a thorough study on a new esterase, expressed in an uncommon host.
More sequence information of the esterase should be provided: what is the most similar known esterase (seq. identity?!).
It would be informative to give an estimate of how much enzyme could be purified from 1 L of culture.
Kcat should be kcat
Do not report determined values with a suggested accuracy that cannot be true: for example, 257.7 is a 4-digit number suggesting that the measurement allows such accurate determination. This is wrong.
The authors should use an AlphaFold2.0-predicted structure. It does not make sense to work with a model based on a His-tagged sequence. And homology modeling should be prevented in this era of AI.
Comments on the Quality of English LanguageWhy Haloferax abbreviated to Hfx.??
Abbreviate every organism name after the first mentioning! This goes wrong several times.
‘determinate’?
Glycosylation > glycosylation
Author Response
Dear Reviewer 2: We deeply appreciate your comprehensive and detailed review of our paper. We appreciate your great contribution to have a better publication.
Questions to reply:
1) More sequence information of the esterase should be provided: what is the most similar known esterase (seq. identity?!).
Response: There are many sequences of Halobacterium species with high identity to the esterase of Halobacterium salinarum NCR-1 (HsEst); However, to our knowledge, all these esterases have not been expressed, purified and characterized. The recombinant esterase (rHsEst) presented in this work is the first esterase of Halobacterium salinarum NCR-1 cloned and characterized.
The only characterized esterases (from other organisms), in which a structure is deposited (sequences with _* in Fig. S1), have the lower similarity with HsEst. To further emphasize this fact, a supplementary figure (Fig. S1) was added, that is cited and argued in the article, making changes in sections 2.3, 3.1 and 3.2.
Additional information from the NCBI server blasts is attached for the reviewer in the NCBI Blast_HsEst pdf file.
2) It would be informative to give an estimate of how much enzyme could be purified from 1 L of culture.
Response: Enzyme production was 70 U L-1 of culture. This information was included in section 3.3, line 336.
3) Kcat should be kcat
Response: KCAT has been replaced by kcat throughout the document.
4) Do not report determined values with a suggested accuracy that cannot be true: for example, 257.7 is a 4-digit number suggesting that the measurement allows such accurate determination. This is wrong.
Response: The reported values were modified according to your suggestion.
5) The authors should use an AlphaFold2.0-predicted structure. It does not make sense to work with a model based on a His-tagged sequence. And homology modeling should be prevented in this era of AI.
Response: It is true that Alphafold's attempt to solve the structure of several protein families has been a success; However, this does not apply to all proteins, as may be the case of our esterase (HsEst), of which there are not many references and a structure obtained by any method of an enzyme similar to HsEst (e.g. X-ray diffraction, electron cryomicroscopy , NMR of proteins) compared with the model obtained with AlphaFold could help verify this. Despite this, the model of rHsEst was also carried out with AlphaFold2 CoLab and compared with the one obtained with Phyre2. To further emphasize this fact, supplementary figures (Fig. S4, S5, S6 and S7) were added, that are cited and argued in the article, making changes in sections 2.3, 3.1 and 3.2.
Comments on the Quality of English Language
- Why Haloferax abbreviated to Hfx.??
Response: Abbreviation Hfx. was changed to H. throughout the document.
- Abbreviate every organism name after the first mentioning! This goes wrong several times.
Response: Names abbreviations of each organism were corrected in the document.
- ‘determinate’?
Response: This mistake was corrected in the document.
- Glycosylation > glycosylation
Response: “Glycosilation” was corrected in the document

Reviewer 3 Report
Comments and Suggestions for Authors
I regret to inform the authors that I must recommend the rejection of this manuscript. The document suffers from numerous issues with its writing quality and exhibits significant redundancy in both the results and methods sections. It is very difficult to maintain the flow of text. Furthermore, it has been noted that the study's contribution is just incremental and has already been reported in similar enzymes from other organisms. Below are the just few specific points highlighting some minor concerns:
Material and Methods:
-
1. The manuscript lacks clarification regarding the abbreviations "Hv-Ca" and "Hv-YPC medium." There is no mention of their full forms or compositions, making it challenging for readers to understand the experimental setup. Moreover, the study's contribution is reported as incremental, which necessitates a robust justification for conducting the research in the first place.
-
2. The choice of measuring bacterial optical density (OD) at 650 nm, as opposed to the conventional 600 nm, is not explained. Authors should provide a clear rationale for this deviation from the standard method. Given the incremental nature of the study, such explanations become even more critical.
-
3. The sentence "Purified rHsEst was exposed to various chemical species under the following experimental conditions: NaCl 1 M, rHsEst esterase activity 1.99 ± 0.04 U mL-1, rHsEst 0.37 ± 0.06 mg L-1, and incubation at 30°C for 1 h" is convoluted and challenging to understand. The authors should rephrase and clarify this section for better comprehension, especially in light of the incremental contribution.
-
4. Mention of "kinetic assay plots" is made, but the actual plots are missing in the manuscript or supplementary materials. Given the incremental nature of the study, including these plots is crucial for readers to assess the findings more effectively.
Results:
- 1. The statement on line 458, regarding rHsEst's specificity for shorter chain NPCs, is unclear and lacks proper explanation. Authors should elaborate on the reasons behind the reduced activity for pNPC2. Additionally, the manuscript should explicitly address how this finding differs from previous research on similar enzymes from other organisms.
In conclusion, the manuscript not only suffers from issues with clarity, completeness, and writing quality but also lacks a clear justification for its incremental contribution, considering similar research on enzymes from other organisms. The redundancy in the manuscript needs to be significantly reduced. If the authors can make the necessary revisions and provide a robust rationale for their research, the manuscript may be reconsidered for further review. However, as it stands, I cannot recommend its acceptance in its current form.
